# Genes Modulating the Increase in Sexuality in the Facultative Diplosporous Grass *Eragrostis curvula* under Water Stress Conditions

**DOI:** 10.3390/genes11090969

**Published:** 2020-08-21

**Authors:** Juan Pablo Selva, Diego Zappacosta, José Carballo, Juan Manuel Rodrigo, Andrés Bellido, Cristian Andrés Gallo, Jimena Gallardo, Viviana Echenique

**Affiliations:** 1Centro de Recursos Naturales Renovables de la Zona Semiárida (CERZOS—CCT—CONICET Bahía Blanca), Camino de la Carrindanga km 7, Bahía Blanca 8000, Argentina; jpselva@criba.edu.ar (J.P.S.); dczappa@criba.edu.ar (D.Z.); jcarballo@cerzos-conicet.gob.ar (J.C.); juanmanuelrodrigo@outlook.com (J.M.R.); andresmbellido@gmail.com (A.B.); gallo.cristian.andres@gmail.com (C.A.G.); jgallardo@cerzos-conicet.gob.ar (J.G.); 2Departamento de Agronomía, Universidad Nacional del Sur (UNS), San Andrés 800, Bahía Blanca 8000, Argentina

**Keywords:** weeping lovegrass, apomixis, drought stress, RNA-seq, differentially expressed genes, sexuality

## Abstract

*Eragrostis curvula* presents mainly facultative genotypes that reproduce by diplosporous apomixis, retaining a percentage of sexual pistils that increase under drought and other stressful situations, indicating that some regulators activated by stress could be affecting the apomixis/sexual switch. Water stress experiments were performed in order to associate the increase in sexual embryo sacs with the differential expression of genes in a facultative apomictic cultivar using cytoembryology and RNA sequencing. The percentage of sexual embryo sacs increased from 4 to 24% and 501 out of the 201,011 transcripts were differentially expressed (DE) between control and stressed plants. DE transcripts were compared with previous transcriptomes where apomictic and sexual genotypes were contrasted. The results point as candidates to transcripts related to methylation, ubiquitination, hormone and signal transduction pathways, transcription regulation and cell wall biosynthesis, some acting as a general response to stress and some that are specific to the reproductive mode. We suggest that a DNA glycosylase EcROS1-like could be demethylating, thus de-repressing a gene or genes involved in the sexuality pathways. Many of the other DE transcripts could be part of a complex mechanism that regulates apomixis and sexuality in this grass, the ones in the intersection between control/stress and apo/sex being the strongest candidates.

## 1. Introduction

Apomixis refers to asexual propagation by seeds and it is a process composed of three components: apomeiosis, parthenogenesis and autonomous endosperm development or pseudogamy. Three hundred out of more than 400 species of angiosperms that reproduce by apomixis occur in the Poaceae, Asteraceae or Rosaceae [1]. However, some capacity for sexuality is usually maintained; thus, they benefit from using a very sophisticated combination of reproductive strategies, generating diversity and, concurrently, allowing the best fitted individuals to propagate clonally [2]. Due to its polyphyletic origin, there are numerous forms of apomixis: diplospory, apospory and adventitious embryogenesis [3]. The elucidation of the mechanism of apomixis is important not only for biological interest, but also for agricultural technology. In the agricultural industry, it is predicted that apomixis would decrease the cost of hybrid seed production significantly and increase the yield of existing inbred crops by converting them into high-yielding hybrids [4].

Although several comparative transcriptomic studies have already been performed in apomictic species, such as *Pennisetum ciliare* [5], *Brachiaria brizantha* [6], *Poa pratensis* [7], *Panicum maximum* [8], *Paspalum simplex* [9], *Hieracium praealtum* [10], *Ranunculus auricomus* [11], *Boechera gunnisoniana* [12] and *Hypericum perforatum* [13,14], and several candidate genes triggering specific components of apomixis are known, the function of their proteins is still not clear. Among the genes associated with the components of apomixis, found both in studies of apomictic species and in mutants that resemble apomixis, the following can be mentioned: *SERK* and *APOSTART* [15], *DIF1* [16], *BABY BOOM* [17], *APOLLO* [18], *DEMETER* [19], *MSII* [20], *RDR6* and *SGS3* [21], *DYAD/SWITCH1* [22,23], *AGO9* [24], *AGO104* [25], *ORC* [26], *GID1* [27], *FIE* [28], *AGAMOUS-LIKE 62* [29], *PnTgs1-like* [30] and *DMC1* [31].

In organisms that can reproduce both sexually and asexually (facultative apomictic), stress plays an important role in determining which reproductive mode is used [32]. For example, in species with cyclical apomixis (like *Daphnia pulex*), reproduction is apomictic in one season, generally under favorable conditions, and sexual during stressful conditions [33]. Different species of fungi, algae and insects induce sexual reproduction under unfavorable conditions or in response to abiotic stress [34,35,36]. In plants, abiotic stress, such as drought or heat, can induce megaspore mother cells (MMCs) to undergo meiosis in the ovules of apomictic plants and produce genetically-reduced (sexual) embryo sacs [37,38,39,40]. Drought- and heat-stressed *Boechera lignifera* and *B. gunnisoniana* achieved major shifts from apomeiosis to meiosis in MMCs, whereas the non-stressed control plants exhibited 87–98% apomictic dyad formation, the heat- and water-stressed plants exhibited 75–80% sexual tetrad formation [38]. Differences in the photoperiod also induce increases in sexuality, such as in *Themeda australis* [41] and *Ranunculus auricomus* under long photoperiods [39] and in *Paspalum cromyorrhizon* [42] and *Brachiaria brizantha* [43] under short day conditions. The reproductive modes of other species also respond to environmental conditions; for instance, Gounaris et al. [44] detected a greater number of reduced embryo sacs under salt stress in *Cenchrus ciliaris*. Based on this, it is reasonable to expect that facultative apomicts tend to switch to sexual reproduction more often under stress conditions, and that such a stress-dependent switch facilitates the organism’s adaptation to a stressful environment [40].

Few studies associate the change in the frequency of apomictic/sexual embryo sacs under stress conditions with changes in gene expression. RNA-Seq studies conducted with immature pistils taken from drought-stressed and well-watered sexual and apomictic *Boechera* spp. plants show that this stress-induced switching includes global epigenetic-based changes in gene expression [45]. Gene ontology (GO) analyses of these differences in gene expression indicate that oxidative stress induces meiosis to occur instead of apomeiosis in apomictic *Boechera* [45]. Several authors, in different model plants, found that the genes that participate in stress pathways are related to the determination or regulation of apomixis [13,39,40].

*Eragrostis curvula* (weeping lovegrass), an African grass with cytotypes of different ploidy levels (e.g., 2x–8x) and displaying obligate and facultative apomixis and sexual reproduction [46], has become a model for the analysis of apomixis mechanisms, due to its particular diplosporous development (meiotic diplospory maintaining the same embryo: endosperm ploidy ratio as in sexual seeds). In recent years, the reproductive mode of this grass was studied extensively, providing information about the cytoembryological aspects of its apomictic–sexual development [47], differentially expressed (DE) transcripts [48,49,50,51], epigenetic aspects of apomictic regulation [52,53], mapping of the apomixis locus [54] and a high quality genome assembly [55]. Our group also demonstrated that under different internal and external stressful situations, including a change in ploidy, water stress, in vitro culture and intraspecific hybridization, the number of sexual embryo sacs increased in facultative apomictic plants of this grass [40,56]. Our group was able to observe that plants of the Tanganyika INTA cultivar produced fewer than 2% of sexual embryo sacs when growing in optimal conditions, but under different stress situations these plants showed an increase in the number of sexual embryo sacs [40]. This increase in the level of sexuality was associated with genetic and epigenetic changes, like methylation and molecular markers [40,56]. Evidence of epigenetic mechanisms controlling apomixis was also observed since differentially expressed patterns of RNA-directed DNA methylation (RdDM) genes [52] and microRNA between sexual and apomictic genotypes [53] were recently reported in this grass.

The aim of the present study was to identify genes that are differentially expressed in weeping lovegrass inflorescences of control and water-stressed facultative apomictic plants and to compare them with the differentially expressed genes between apomictic and sexual plants previously reported by our group [51]. This approach was taken based on previous findings about increases of sexual processes under stress conditions [40], in order to look for common pathways between stress and apomixis. It could give clues about the regulation of this intriguing reproductive mode.

## 2. Materials and Methods

### 2.1. Plant Materials

Plants of the tetraploid (2n = 4x = 40) facultative apomictic Don Walter cultivar were grown in 10 l pots in the greenhouse. Three plants coming from the same seed set (apomictic background) were divided asexually (two tillers each) and one tiller was assigned to the control treatment and the other one to the stress condition, totaling six plants. To avoid the noise represented by the genotype effect in a high heterozygous grass, we did not include a sexual or a full apomictic genotype as a control. For this reason we worked with clonal plants.

### 2.2. Stress Treatments

Plants were exposed to water stress conditions by water deprivation from three months before the onset of flowering until the end of the flowering season (September to March, 2016–2017). In order to maintain the biological functions of the plants, they were watered weekly with 50–80 mL per pot and a supplementary irrigation of an extra 100 mL per pot was carried out to induce the flowering close to the flowering season. Plants grown under normal conditions (300–500 mL water/week) were used as controls. As an indicator of the plant water status, the relative water content (RWC) was determined at the inflorescence collection time in leaves using the following formula: RWC (%) = (FM−DM)/(TM−DM) × 100, where FM, DM and TM are the fresh, dry, and turgid tissue weights, respectively [57].

### 2.3. Embryo Sacs Analyses

To analyze the effect of water stress on the reproductive mode, the different stages of megasporogenesis and megagametogenesis were observed under an optical microscope. Inflorescences from the control and treated plants were collected at the beginning of anthesis, when all embryo sac developmental stages were observable [47] and they were fixed in FAA (50% ethanol, 5% acetic acid, 10% formaldehyde in distilled water). Then, individual spikelets were dehydrated in a tertiary butyl alcohol series and embedded in Paraplast [58]. Samples were sectioned at 10 μm and stained with safranin-fast green. Observations were carried out with a Nikon Eclipse TE300 light transmission microscope. To assess the reproductive mode, the presence of meiosis or the number and position of nuclei in the embryo sac were observed according to Meier et al. [47]. More than three hundred spikelets were observed (41 from control plants and 271 from stressed ones).

### 2.4. RNA Extraction and Sequencing

Spikelets with basal flowers at the beginning of anthesis, containing embryo sacs at all developmental stages, were collected from control (DWC1, DWC2 and DWC3) and treated plants (DWS1, DWS2 and DWS3). In total, 30 mg of fresh tissue from each sample were ground to a fine powder using liquid nitrogen. The total RNA was extracted from the plant tissue as two fractions, small and large RNA, including RNA sequences smaller and larger than 200 bp, respectively, using a commercial NucleoSpin^®^ miRNA kit (Macherey-Nagel, Düren, Germany) according to the manufacturer’s instructions. The large RNA fraction was sequenced in 150 bp reads in pair-end through an Illumina HiSeq1500 platform at INDEAR (Rosario, Argentina).

### 2.5. Bioinformatics Analyses

Quality assessments of the reads were performed using the FastQC software. Subsequently, the reads were filtered using the Trimmomatic software [59] with the following parameters: ILLUMINACLIP = TruSeq3-PE-2.fa:2:30:7:4:false, LEADING = 3, TRAILING = 3, SLIDINGWINDOW = 4 = 20, CROP = 150, HEADCROP = 13 and MINLEN = 36. The resulting paired-end reads were assembled using the Trinity software [60] with a KMER_SIZE:32. In order to remove redundant transcripts, a clustering with a sequence identity threshold of 0.9 was performed using the CD-HIT software [61,62]. A quality assessment of the assembly was made using the standard metrics provided by the downstream analysis of Trinity [63] and BUSCO [64,65].

Regarding the differential expression analysis, the transcript quantification (estimation of the abundance of each assembled transcript) was performed with RSEM software [66] using the trimmed paired and unpaired reads aligned by Bowtie2 as an input [67] with the parameters: fragment_length = 137 and fragment_std = 23. The differential expression analysis was carried out using the EDGE R-Cran package [68,69] and the DE transcripts were selected using a fold change (FC) = 2 and an *e*-value of 1e-3.

A functional annotation of the DE genes was performed with Blast2GO [70]. The distribution of level 2 and 3 GO terms—including biological process, cellular component and molecular function among the DE annotated transcripts—were plotted with Blast2GO. A comparative analysis of the GO terms containing the annotated down- and upregulated transcripts under stress conditions was performed and plotted on a bar chart. The KEGG pathways (Kyoto Encyclopedia of Genes and Genomes [71]) were also compared to detect differentially enriched pathways between the control and treated plants.

The Heatmap analysis was performed using the R Package pheatmap [68] using as an input the expression matrix used for the differential expression analysis.

### 2.6. Comparison with Previously Sequenced E. curvula Transcriptomes

The DE transcripts between the control and stressed plants and the DE transcripts between sexual and apomictic plants previously obtained [51] were compared in order to find a relationship between the genes/pathways related to stress/increase in sexuality and the genes/pathways involved in apomixis/sexuality using a unidirectional exonerate [72] alignment with a minimum identity of 90% and minimum coverage of 50% to match the sequences.

### 2.7. Search for Long Noncoding RNAs

Detection of long noncoding RNA (lncRNA) was carried out using the DE transcripts that could not be annotated with the Blast2GO software using the Magnoliopsida nonredundant protein database. The Coding Potential Calculator (CPC) software [73] was run using these transcripts as input, with the default parameters to detect the potential coding and lncRNA sequences. Finally, the lncRNA sequences annotated with the CPC software were searched in the dataset of DE transcripts between sexual and apomictic plants [51] in order to identify common transcript sequences.

### 2.8. Validation of Gene Expression by Quantitative Real-Time PCR (qRT-PCR)

RNA was extracted as detailed above, under the same conditions and using the same RNA extraction kit used for the Illumina sequencing. The cDNA synthesis was performed using the ImProm II Reverse Transcription System (Promega) following the supplier’s instructions. The cDNA amplification was performed using specific primers designed according to the Integrated DNA technology (IDT) webpage (https://www.idtdna.com/scitools/Applications/RealTimePCR/). The primer pairs used in the qRT-PCR experiments are shown in Appendix A. Real-time PCR reactions included 50 pmol of forward and reverse primers, 5 μL of cDNA diluted 100-fold and 10 μL of Real Mix (Biodynamics, Argentina). The amplification was carried out in a Rotor Gene 6000 thermocycler (Corbett Research, Australia). The expression level was normalized against the *E. curvula* UBICE gene (Appendix A). The thermal cycling used for amplifications was as follows: 95 °C for 2 min, followed by 45 cycles at 94 °C for 10 s, then 15 s at the optimal annealing temperature for each primer pair and finally at 72 °C for 20 s using three biological and three technical replicates. The specificity of each reaction was verified through the dissociation curve profiles. To calculate the relative expression level and primer efficiency estimation, background-corrected raw fluorescence data were imported into LinRegPCR software version 11.0 [74,75]. The program uses a linear regression analysis to fit a straight line and estimate the PCR efficiency of each individual sample based on the slope of this line. The statistical analysis of the qRT-PCR fold change in the expression of genes among different treatments was analyzed using a Student’s t Test. A *p*-value of 0.05 was considered to be significant.

### 2.9. Data Availability

The Transcriptome Shotgun Assembly project has been deposited at DDBJ/EMBL/GenBank under the accession GIQX00000000. The version described in this paper is the first version GIQX01000000.

## 3. Results

### 3.1. Percentage of Sexual vs. Apomictic Processes in Control and Water-Stressed Plants

The number of pistils and the percentages of sexual and apomictic embryo sacs from the control and water-stressed plants are shown in Table 1. The cytoembryology analyses showed that 4.97% of the embryo sacs were sexual (*n* = 161) in control plants, with an average RWC of 81.9%. Under water-stressed conditions, with an average RWC of 49.7%, the percentage of sexual embryo sacs increased to 23.84% (*n* = 172). The spikelets from stressed plants were smaller and had fewer flowers than the control ones. They also had a higher number of aborted seeds, so it was necessary to observe more spikelets to get an *n* = 172.

### 3.2. Sequencing and Assembly

Six RNA TruSeq HiSeq1500 Illumina libraries, corresponding to three biological replicates of control plants and three from water-stressed plants, were sequenced producing a total of 172,128,258 paired-end reads (2 × 150 bp). Then, after the quality analysis, performed with FastQC [76], the reads were trimmed resulting in 95,127,754 paired-end reads. These paired-end reads were de novo assembled (Trinity software) resulting in 305,798 transcripts. Finally, the redundant transcripts were removed and the remaining transcripts were clustered using the CDHIT software, resulting in a final number of 201,011.

Although an *E. curvula* genome assembly is available [55] and we could have used it as a reference genome, for this study we decided to de novo assemble the reads since the sequenced genome belongs to a sexual diploid genotype and the sequences corresponding to the region that determine apomixis might not be present, as was observed by Zappacosta et al. [54] using molecular markers linked to the trait. The assembly quality analysis, performed with Trinity stats, indicated an N50 of 1553 bp, an average contig length of 919.09 bp and a GC content of 46.38%. The percentage of aligned reads, determined by the Bowtie2 software, was 96.45% and the percentage of complete BUSCO genes in the final assembly was 87.2% (S = 54.6%, D = 32.6%), with 9.6% of fragmented and 3.2% of missing genes.

### 3.3. Differentially Expressed Transcript Analysis

A total of 501 DE transcripts were obtained with a fold change of two and an *e*-value of 1e-3. Out of these, 350 were downregulated while 151 were upregulated in the stressed plants. Appendix A shows all the information about the DE transcripts (ID and Blast, Blast2GO and KEGG analyses).

A principal component analysis (PCA) computed on the DE transcripts effectively separated the control from the treated samples, with the two first principal components explaining approximately 90% of the overall variance (Appendix A).

### 3.4. Gene Ontology Analysis

The gene ontology classification made in order to identify the pathways potentially associated with the increase in sexual pistils under stress conditions gave a result of 380 out of 501 DE annotated transcripts. This classification (Figure 1) shows that the transcripts upregulated and downregulated under stress treatments were included in the same main categories (except the rhythmic process and growth). However, the number of GO terms with downregulated transcripts was higher (264 downregulated vs. 116 upregulated under stress). This effect could be part of the general decrease in gene expression that happens under stress conditions due the lack of resources.

To carry out a more specific analysis, we searched for the presence of DE transcripts in the GO terms reproduction and reproductive processes. We found two upregulated DE transcripts with homology to a G-type lectin S-receptor-like serine/threonine-protein kinase SD2-5 GsSRK (DN35954_c0_g1_i7) and *grassy tillers1* (DN35816_c1_g1_i1) and five downregulated DE transcripts under stress with homology to stromal processing peptidase (DN36893_c0_g1_i3), β-expansin (DN35576_c3_g4_i12, DN35576_c3_g4_i9), B3 domain-containing protein LFL1 (DN37977_c0_g1_i3) and indole-3-pyruvate monooxygenase YUCCA2 (DN36728_c1_g2_i3). Although some of these genes are related to stress responses, such as GsSRK, the overexpression of this gene in *Arabidopsis* exhibited more siliques at the adult developmental stage, among other traits [77]. All these DE transcripts are present in pathways that could participate in reproduction regulation, since they are involved in auxin pathways (YUCCA2, [9]), transcription regulation (*grassy tillers1*, [78]), embryogenesis (stromal processing peptidase, [79]), flowering time regulation (LFL1, [80]) and cell wall biosynthesis and pollen tube penetration through the stigma (β-expansin, [81]).

The GO terms only represented by upregulated transcripts at level 3 (Appendix A) were: chemical response, cell wall organization or biogenesis, response to biotic stimulus, response to external stimulus, response to other organisms, drug binding, cofactor binding, ligase activity, catalytic activity (acting on RNA), oxygen carrier activity, carbohydrate binding, envelope, RNA polymerase complex, thylakoid and Sm-like protein family complex. Although several of these GO terms may play a role in determining or regulating apomixis, we would like to highlight the ligase activity, where one of the differential transcripts shows homology with a putative E3 ubiquitin-protein ligase RING1a, an enzyme cited by other authors as a candidate to be involved in apomixis [82].

The GO level 3 terms that were only represented by downregulated transcripts (Appendix A) were: regulation of biological quality, regulation of molecular function, cell communication, signal transduction, response to abiotic stimulus, lyase activity, lipid binding, signaling receptor activity, isomerase activity, quaternary ammonium group binding, structural constituent of ribosome, membrane protein complex, extracellular space, external encapsulating structure and the microtubule associated complex. Among these GO terms, it can be highlighted that the membrane protein complex term contains two transcripts with homology to the AP-1 complex subunit sigma-1 and ENTH domain-containing protein, which were associated with SCD1-mediated vesicle transport and protein ubiquitination, a pathway cited by other authors as related to apomixis [83].

### 3.5. KEGG Pathway Classification

To identify the additional levels of regulation acting on the apomixis/sexuality switch under stress, the DE transcripts were analyzed using the KEGG database. Using the Blast2GO software, 13 transcripts that were upregulated and 37 that were downregulated under stress conditions were assigned to 48 different pathways (Figure 2).

In eleven pathways, DE transcripts from both categories, up- and downregulated, were present while 16 pathways were composed only by upregulated and 21 by downregulated transcripts under stress conditions. The pathways with the highest number of downregulated transcripts were in thiamine metabolism and purine metabolism. On the other hand, the pathways with the highest number of upregulated transcripts were the alanine, aspartate and glutamate metabolisms, arginine biosynthesis and the amino sugar and nucleotide sugar metabolisms. The upregulated transcripts included in more pathways were the enzymes aspartate aminotransferase cytoplasmic (ec:2.6.1.1—transaminase, DN38803_c1_g1_i6) and glutamine synthetase cytosolic isozyme 1–3 (ec:6.3.1.2—synthetase, DN38913_c0_g1_i4). The downregulated ones included the enzymes NADPH-dependent aldo–keto reductase (ec:1.1.1.2—dehydrogenase NADP+, DN38835_c1_g1_i3) and phosphoenolpyruvate carboxylase 4 (ec:4.1.1.31—carboxylase, DN19844_c0_g1_i1).

### 3.6. Comparison with Previous E. curvula Transcriptomes

In order to look for an association between the DE transcripts in plants under stress conditions and the increase in the number of sexual pistils, the results obtained in this study were compared with the DE transcripts obtained in a previous study (sexual vs. apomictic [51]), where two transcriptomes of weeping lovegrass, a fully sexual and a fully apomictic one, were sequenced.

Ninety three out of the 501 DE transcripts showed homology in the unidirectional alignment with the 9750 DE transcripts reported by Garbus et al. [51]. Sixty nine out of the 93 DE transcripts belonged to the downregulated in the stress group and 24 to the upregulated one (Figure 3 and Appendix A). From this analysis we can point out two interesting subgroups of transcripts that could be related to the increase in sexuality (Table 2 and Appendix A). The first one is composed of eight transcripts that were downregulated under stress conditions and upregulated in the apomictic genotype (in the apomixis/sexuality comparisons). The other group is composed of 20 transcripts that were upregulated under stress and downregulated in the apomictic genotype (Table 2).

Table 2 shows that there is more than one transcript for the same gene. It is important to remark that the β-expansin and *grassy tillers1* genes were also annotated under the GO terms associated with reproduction. Another result to be highlighted from this analysis is a subtilisin (SBT5.3) downregulated under stress and overexpressed in apomictic plants. A member of this gene family, SBT1.4, was successfully validated by qRT-PCR. Subtilisins are proteases that are associated with the early stages of seed development [84] and reproductive mode [85,86].

### 3.7. Long Noncoding RNAs

From 501 DE transcripts blasted against the Magnoliopsida database, 452 were homologous with a coding sequence whereas 49 did not find a hit (Appendix A). Over these 49 transcripts introduced in the CPC software, 33 were predicted as noncoding with a high confidence. In total, 24 out of the 33 putative lncRNA were downregulated under stress conditions and nine were upregulated, representing 6% of the corresponding categories. Interestingly, 3 out of 33 of these RNAs were also found DE between apomictic and sexual plants in our previous work [51].

### 3.8. Analysis of Differentially Expressed Transcripts

The annotation of the DE transcripts that were not highlighted by the GO terms or KEGG pathways was also analyzed. Table 3 summarizes the annotated DE transcripts up- or downregulated under the stress treatment that have previously been mentioned by different authors in apomictic species or mutants. Among them, we can mention the transcription factor ethylene-responsive AINTEGUMENTA-like 5 (AIL5), belonging to the AP2 family, the same family as BABY BOOM (BBM), one of the few candidate genes for apomixis that has been confirmed to play a role in parthenogenesis [15]. Three DE transcripts were found associated with the brassinosteroid pathway (EXORDIUM, guanine nucleotide-binding protein α-1 subunit and serine/threonine protein phosphatase 2A 55 kDa regulatory subunit B). This pathway has previously been mentioned by other authors as being associated with megagametogenesis in Poaceae [87] and apomixis induction [88].

Another DE transcript upregulated under the stress treatment showed homology with MO25, a protein present in the apomixis-determining region in *Hypericum perforatum* [13] and associated with signal transduction (GO terms: intracellular signal transduction and positive regulation of protein serine/threonine kinase activity).

A very interesting candidate also related to apomixis that was found to be DE (upregulated under stress conditions) in the present study is a transcript with homology to the repressor of silencing ROS1A, a DNA-glycosylase that removes methylated cytosines and replaces them with unmethylated cytosines. If we postulate that sexuality is silenced in facultative apomictic plants, this gene is a very attractive candidate to de-repress this function by a demethylation pathway under stress situations, allowing an increase in the number of sexual pistils.

Among other genes previously related to apomixis and downregulated under stress conditions are the SNF1-related protein kinase regulatory subunit β-1 and F-box proteins (At2g14290, At3g07870, and At5g07610). Regarding genes previously related to apomixis and upregulated under stress, the transcription factor NAC10 should be mentioned.

### 3.9. Validation by qRT-PCR

qRT-PCR assays were used to corroborate the in silico differential expression analysis of key genes (Figure 4a and Appendix A), which were selected on the basis of their expression pattern and/or annotation. Heatmap (Figure 4b) was also used to show the significance of the in silico differential expression of the six selected genes.

EcAIL5-like, EcROS1-like and EcSINA_5-like, a transcription factor that belong to the AP2 family, a DNA glycosylase, and a putative E3 ubiquitin-protein ligase, respectively, were chosen for validation because they have previously been mentioned to be involved in pathways recently identified as having regulatory roles in apomixis.

An interesting transcript among those selected for qRT-PCR validation was EcDN14802, a transcript that was predicted as lncRNA, downregulated under stress conditions and not found in *E. curvula* sexual genotypes (OTA-S transcriptome and Victoria genome). These observations point out this transcript as a possible candidate related to apomictic processes. With the same criteria, the transcript EcNon-LTR-like was also selected. Both transcripts were found only in the apomictic genotype Don Walter.

EcSBT1.4-like is a stress downregulated protease that was selected for validation because it was previously found in *E. curvula* transcriptomes (Table 2) and because it was included in the GO terms reproductive process and reproduction.

Although EcNon-LTR-like and EcROS1-like were not statistically validated by qRT-PCR, they were very close and the Heatmap analysis showed a very consistent expression across all the three samples in each treatment.

## 4. Discussion

Our experimental design was aimed at finding a relationship between stress, the increase in sexual processes and the candidate genes involved using a facultative apomictic cultivar of *E. curvula*. Prior to the analysis, it is important to consider four aspects of the experimental design. Firstly is the fact that the experiments were conducted with clonal plants, so both the control and stressed plants have the same background. Secondly, the treatment involved water stress, hence some of the responses could be exclusively due to stress itself. Thirdly, the biological samples used consisted of whole flowers (spikelets), which comprise raquis, glumes, lemma, palea, ovary and anthers, therefore, the whole set of transcripts characterized here are derived from a variety of cell types, including somatic cells and male and female reproductive cells from premeiosis to anthesis. Finally, both treatments were conducted with a facultative apomictic cultivar, hence we expected to find mostly apomictic/sexual regulators, not apomictic triggers because they must be present in plants from both treatments. On the other hand, we are working with the knowledge that apomixis in *E. curvula* is determined by a genomic region [54] with an epigenetic component modulating the traits [40,52], and apomixis and sexuality co-exist in facultative plants [54], allowing us to hypothesize that sexuality is being repressed at different levels in apomictic plants. This repression is deregulated by different endogenous [56] and exogenous stresses [40], probably mediated by transcriptional and post transcriptional regulatory mechanisms.

In the present study, as in previous ones using this grass, it was shown that stress significantly increased the percentage of sexual processes, as was observed in the facultative apomictic Tanganyika and Don Walter cultivars, where the increase in sexual pistils ranged from 1.8 to 14.4% and from 4 to 22%, respectively [40,56]. Similar results were obtained in this study (4 to 24% increase) using plants of the Don Walter cultivar. This situation has also been observed in other plant species, such as *Boechera*, where Mateo de Arias [38] demonstrated that drought and heat stress caused a shift from apomeiosis to meiosis in female meiocytes when plants are stressed and the frequency of sexual embryo sacs was increased from an average of 10 to 30% in the drought-stress-treated plants, with similar percentages to the ones observed here in *E. curvula*.

Previous studies conducted in sexual and apomictic *E. curvula* flowers using different strategies, like expressed sequence tags (ESTs) [48,49], differential displays [50] and 454 sequencing [51], allowed us to detect that many of the DE transcripts are homologous to genes related with different environmental stresses. This fact, plus the observation that stress increases the frequency of sexual processes, encouraged us to look for an association between both findings. The present study using the Illumina platform represents a deeper and complete transcriptomic sequencing of spikelets from *E. curvula* that provides a great amount of information that can be used to look for DE genes potentially involved in the modulation of apomixis/sexuality. Despite the large number of assembled transcripts (201,011), few were DE between treatments (501). An important proportion of these transcripts could be annotated and classified under the GO terms and KEGG pathways. The nonannotated transcripts (49) were evaluated with the CPC software and 33 were predicted as long noncoding RNAs. The identification of these genes and the lncRNAs are a benchmark for understanding the common pathways between stress and apomixis.

The Blast2GO analysis at level 2 showed that the same GO terms are present in both treatments, having more downregulated than upregulated transcripts in each and this could be due to the overall stress response. An interesting finding was the presence of seven transcripts in the GO terms reproduction and reproductive process that are related to stress responses [77] and were also present in apo/sex comparisons made earlier in *E. curvula* [49], such as GsSRK. Other transcripts included in these terms are involved in auxin pathways (YUCCA2, [9]), transcription regulation (*grassy tillers1*, [78]), embryogenesis (stromal processing peptidase, [79]), flowering time regulation (LFL1, [80]), cell wall biosynthesis and the pollen tube penetration of the stigma (β-expansin, [81]) and could also be associated with apomixis in weeping lovegrass.

Based on homology or annotations, some of the DE transcripts are involved in pathways recently identified as having regulatory roles during apomixis [13,82,83]. One of these pathways is ubiquitination, in which a putative E3 ubiquitin-protein ligase RING1a and a sulfite exporter TauE/SafE family protein 3 were present, among others. Ubiquitination regulates nearly every aspect of cellular events in eukaryotes. It modifies intracellular proteins with the 76-amino acid polypeptide ubiquitin and destines them for proteolysis or activity alteration. The proteasomal degradation-mediated control of key cell cycle regulators and other targets influence reproductive fate decisions and germline development [93,94]. In this study, nine transcripts, representing seven genes that belong to the ubiquitination pathway, were found DE. Five out of these seven genes (E3 ubiquitin-protein ligase SINA-like 5, E3 ubiquitin-protein ligase SINA-like 10, U-box domain-containing protein 35, ubiquitin-NEDD8-like protein RUB2 and U-box domain-containing protein 15) were downregulated under stress conditions and two (putative E3 ubiquitin-protein ligase RING1a and sulfite exporter TauE/SafE family protein 3) were upregulated. This finding provides further evidence that tightly regulated protein degradation affecting cell cycle progression might be of crucial importance for governing the distinct specification and differentiation of apomictic and sexual germlines. Genes related to ubiquitin pathways were also observed in comparisons of apomictic and sexual weeping lovegrass [49,50,51] and *Paspalum notatum* flowers [95] and in ploidy related changes in the apomictic grass *P. notatum* [96].

Together with the ubiquitin pathway, F-box proteins were also mentioned by Zühl et al. [82] as differentially expressed and they might potentially play a role in the reproductive mode. Martelotto et al. [96] found transcripts homologous to F-box genes differentially expressed and related to ploidy changes in the apomictic species *P. notatum*. Here we found six downregulated F-box proteins and one upregulated. F-box proteins are part of Skp1–Cullin1–F-box protein (SCF) ligase complexes, acting in the polyubiquitination-mediated 26S proteasomal degradation [97]. Changes in this pathway due to the stress treatment could also be associated with the apomictic/sexual switch, as in ubiquitination.

Interestingly, when contrasted with previous data, four groups of DE transcripts were also found as DE in the comparison of apomixis vs. sexuality in weeping lovegrass [51]. In Table 2 we highlighted two groups. The first group is composed of eight transcripts that were downregulated under stress conditions and upregulated in the apomictic genotype (in the apo/sex comparisons). The second group is composed of 20 transcripts that were upregulated under stress and downregulated in the apomictic genotype. These 20 transcripts could be the ones that are associated to the increase in sexual embryo sacs. In the first group, transcripts presented homology with subtilisins, β-expansin, pollen allergen Cyn d 15, glycine-rich proteins, guanine nucleotide-binding proteins, pectine esterases inhibitors, cysteine proteases and a long noncoding RNA. These genes are mainly involved in cell wall modifications, proteolysis, signal transduction and a possible regulatory function (lncRNA). In the second group, we can mention transcription factors, such as the NAC domain containing protein, *grassy tillers1* and HOX12; lipoxygenases; tryptamine hydroxycinnamoyl transferases and SNF1-type serine–threonine protein kinases. The latter gene belongs to the SnRK complex and another component of this pathway was found as DE in the present study (MARD1). The SnRK1 kinases control metabolism, growth and development, and stress tolerance by the direct phosphorylation of metabolic enzymes and regulatory proteins and by extensive transcriptional regulation. SnRK1 is also part of a more elaborate metabolic and stress signaling network, which includes the TOR kinase and ABA-signaling. The plant SnRK1–TOR system is heavily intertwined also with hormone signaling pathways, mainly auxins and brassinosteriods. Many components of these complex pathways were found as DE in the present study, and although they are related to stress responses, they may also be involved in the apo–sex ratio change. Mateo de Arias [38] also finds the DE components of these pathways and Gao [89] proposes that the differences in gene expression between apomictic and sexual plants might be triggered by the presence or absence of stress signaling and showed strong evidence that the TOR–brassinosteroids molecular stress response pathway is involved in the apomeiosis/meiosis switch. Recently, Carman et al. [91] patented components of these pathways as apomixis related, which reinforced the involvement of these genes, not only in the stress response, but also in the reproductive mode regulation.

Several genes related to hormones, such as auxins and brassinosteriods, could also be acting, since a number of genes involved in hormone perception and homeostasis, including cytokinins, auxins, and brassinosteroids, were DE in apomictic pistils [9,85,88]. Auxins and brassinosteroids are pathways that respond to stress and are considered to be key components for cell dedifferentiation (totipotence phenomenon) related to somatic embryogenesis [98], a situation that is similar to apomixis.

Another transcript that is overexpressed under stress conditions in the present study is EcROS1-like, a DNA glycosylase with a high homology to ROS1 (repressor of silencing 1) a DNA demethylase that is indispensable in both male and female gametophyte development [90]. The mutation of this gene in *Arabidopsis* showed DNA hypermethylation (an increased level of methylated cytosine) at nearly 5,000 loci involved in different pathways [99]. We suggest that EcROS1-like, as a specific or an unspecific stress response, could be demethylating a key target, thus de-repressing some gene or genes involved in sexuality pathways that are silenced in apomictic plants. Demethylation together with other pathways found here as DE could be part of the complex mechanism that regulates apomixis and sexuality in this grass, the ones in the intersection between control/stress and apo/sex being the strongest candidates.

In general, our results mainly point to the involvement of demethylation, protein degradation, transcriptional and post transcriptional regulatory mechanisms and regulation by plant hormones and signal transduction in the apomixis/sexuality switch regulation under stress situations.

## 5. Conclusions

These data, together with previous results [40], reinforce our previous hypothesis related to apomixis regulation in weeping lovegrass and its connections with stress. In this model, both pathways—the apomictic and the sexual one—co-exist in facultative apomictic plants, the sexuality mainly being repressed or expressed at very low levels under normal conditions. Stress situations can de-repress sexuality and the number of sexual embryo sacs increases in apomictic plants. We suggest that EcROS1-like can be demethylating, thus de-repressing some gene or genes involved in the sexuality pathways. Many of the other transcripts found as DE could be part of the complex mechanism that regulate apomixis and sexuality in this grass, the ones in the intersection between control/stress and apo/sex being the strongest candidates. Probably some of them are being upregulated under stress by this general demethylation response. Other related processes involved are ubiquitination, hormone and signal transduction pathways, transcription regulation and cell wall biosynthesis, some acting as a general response to stress and some that can be specific to the reproductive mode.

Finally, the availability of the sequence database reported here would make possible the characterization and validation of the numerous genes involved in apomixis and stress.

## Figures and Tables

**Figure 1 genes-11-00969-f001:**
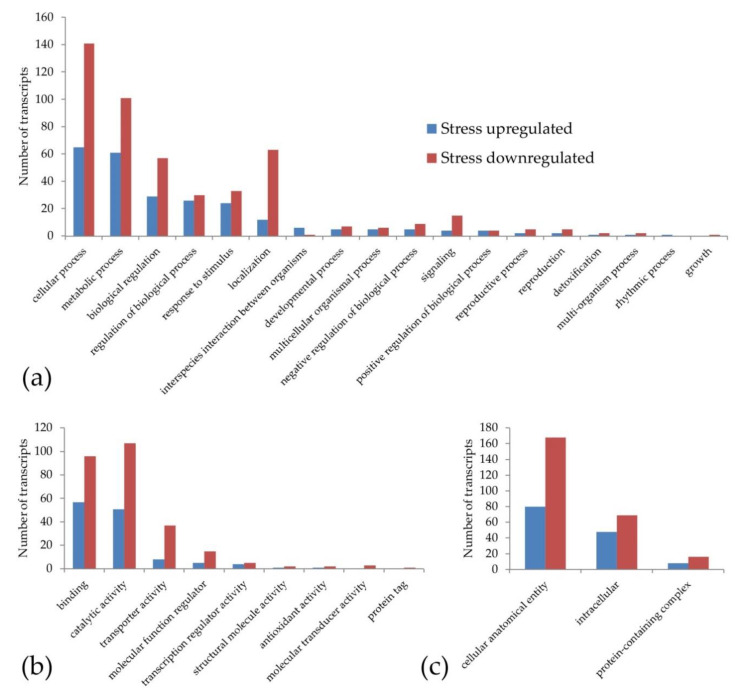
Classification of the weeping lovegrass differentially expressed transcripts according to gene ontology: (**a**) biological process (**b**) molecular function and (**c**) cellular component. Each main category was classified at level 2. Blue bars show the transcripts that are upregulated and red bars represent the transcripts that are downregulated under stress conditions.

**Figure 2 genes-11-00969-f002:**
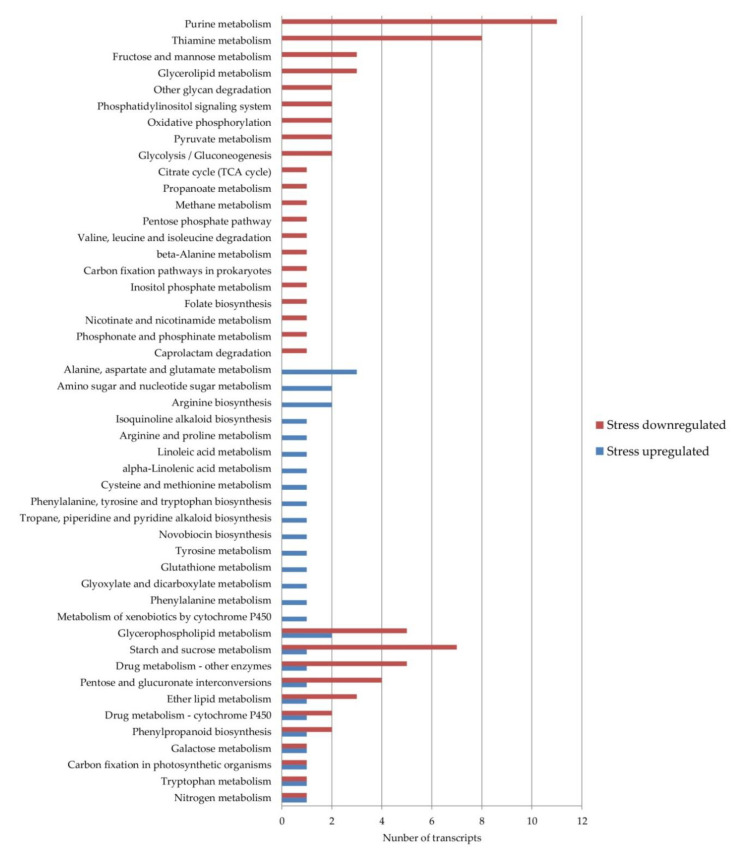
Weeping lovegrass differentially expressed the transcripts present in Kyoto Encyclopedia of Genes and Genomes (KEGG) pathways. The blue bars show the number of transcripts upregulated and the red bars show the number of transcripts downregulated under stress conditions in each pathway.

**Figure 3 genes-11-00969-f003:**
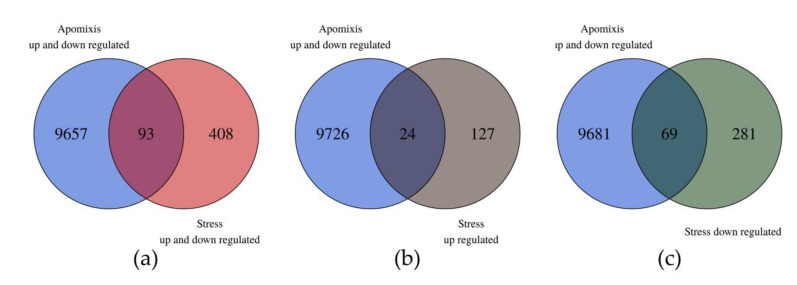
Venn diagrams comparing the weeping lovegrass differentially expressed (DE) transcripts between control vs. stressed plants with the DE transcripts between apomictic vs. sexual plants. Data for the comparison apo/sex come from Garbus et al. [51]. (**a**) In the intersection between both groups of transcripts, there are 93 in common; (**b**) in the intersection are the common transcripts that are DE in apomictic plants (up- and downregulated) and downregulated under stress conditions; (**c**) in the intersection are the common transcripts that are DE in apomictic plants (up- and downregulated) and upregulated under stress conditions.

**Figure 4 genes-11-00969-f004:**
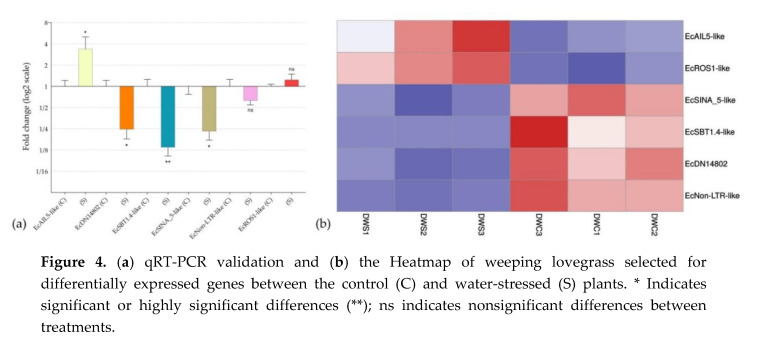
(**a**) qRT-PCR validation and (**b**) the Heatmap of weeping lovegrass selected for differentially expressed genes between the control (C) and water-stressed (S) plants. * Indicates significant or highly significant differences (**); ns indicates nonsignificant differences between treatments.

**Table 1 genes-11-00969-t001:** Percentage of sexual and apomictic embryo sacs in the control and water-stressed *E. curvula* plants (Don Walter cultivar) and the relative water content (RWC) for each treatment.

Embryo Sacs (%)			
Control (*n* = 161)		Stress (*n* = 172)	
Apo	Sex	Apo	Sex
95.03%	4.97%	76.16%	23.84%
(RWC = 81.9 ± 4.1)		(RWC = 49.7 ± 0.6)	

**Table 2 genes-11-00969-t002:** Differentially expressed transcripts between the control and stressed weeping lovegrass flowers showing homology with DE genes between apomixis/sexual flowers obtained previously in *E. curvula* by Garbus et al. [51]. In the description column the annotation provided by Blast2go was included.

Stress Downregulated/Apomixis Upregulated	
Transcript ID	Description
DN38408_c2_g4_i2	subtilisin-like protease SBT5.3
DN35576_c3_g4_i12	β-expansin
DN38000_c1_g1_i5	pollen allergen Cyn d 15
DN25763_c0_g1_i3	Non-annotated
DN40203_c3_g2_i3	fibroin heavy chain
DN25402_c0_g2_i1	cysteine protease
DN33014_c0_g2_i2	guanine nucleotide-binding protein α-1 subunit
DN36051_c2_g3_i1	pectinesterase inhibitor 10
**Stress Upregulated/Apomixis Downregulated**	
DN37495_c2_g2_i1	hypothetical protein EJB05_27044, partial
DN35806_c0_g1_i1	stem-specific protein TSJT1
DN17975_c0_g1_i2	predicted protein
DN39515_c0_g1_i11	NAC domain-containing protein 110
DN39515_c0_g1_i3	NAC domain-containing protein 110
DN39515_c0_g1_i4	NAC domain-containing protein 110
DN33023_c1_g1_i3	linoleate 9S-lipoxygenase 2
DN40610_c1_g1_i1	lipoxygenase 1.1
DN40610_c2_g3_i2	linoleate 9S-lipoxygenase 2
DN30943_c0_g1_i1	tryptamine hydroxycinnamoyltransferase 2
DN30943_c0_g3_i1	tryptamine hydroxycinnamoyltransferase 2
DN35816_c1_g1_i1	*grassy tillers1*
DN35816_c1_g1_i2	homeobox-leucine zipper protein HOX12
DN35242_c1_g4_i1	SNF1-type serine–threonine protein kinase
DN36533_c0_g1_i4	xylanase inhibitor protein 1-like
DN36533_c0_g1_i5	xylanase inhibitor protein 1-like
DN36533_c0_g3_i5	xylanase inhibitor protein 1-like
DN36484_c3_g4_i1	dormancy-associated protein 1
DN37042_c3_g3_i9	dormancy-associated protein 1
DN27261_c2_g1_i1	pathogenesis-related protein PR-4-like

**Table 3 genes-11-00969-t003:** Differentially expressed transcripts reported by different authors in different apomictic species compared with their sexual counterparts.

SeqName	Description	Condition	Probable Function	Reference
DN32086_c1_g2_i1	AP2-like ethylene-responsive transcription factor AIL5	Up	Transcription factor	[15]
DN30159_c1_g5_i1	protein EXORDIUM	Up	Brassinosteroid pathway	[89]
DN37585_c0_g1_i12	serine/threonine protein phosphatase 2A 55 kDa regulatory subunit B β isoform	Down	Brassinosteroid pathway	[89]
DN36281_c2_g1_i5	F-box protein	Down	F-box	[82]
DN36248_c0_g1_i2	F-box protein	Down	F-box	[82]
DN30985_c0_g1_i9	F-box protein At5g07610-like	Down	F-box	[82]
DN23587_c0_g1_i1	F-box domain containing protein	Up	F-box	[82]
DN36607_c0_g1_i5	protein ROS1	Up	Transcription factor	[90]
DN34979_c0_g1_i2	putative MO25-like protein At5g47540	Down	Signaling	[13]
DN30222_c0_g5_i1	transducin/WD40 repeat-like superfamily protein	Down	Histone binding	[38]
DN30614_c2_g1_i7	protein MARD1	Up	SnRK1 regulation pathway	[91]
DN29708_c0_g2_i2	putative vesicle-associated membrane protein 726	Down	Vesicle-mediated transport	[83]
DN27157_c0_g2_i1	LIM domain-containing protein PLIM2b	Down	Actin filament binding	[92]
DN38850_c0_g1_i10	Methylesterase 17	Down	Auxin pathway	[9]
DN36845_c1_g1_i17	kinesin-like protein KIN-14D isoform X1	Down	Microtubule, mitosis	[85]
DN35282_c0_g2_i2	zinc finger CCCH domain-containing protein 35	Up	RNA processing, cell cycle	[38]
DN38610_c0_g1_i2	Tyrosine-protein kinase BAZ1B	Up	Chromatin remodeling	[12]
DN32750_c0_g1_i2	protein TONSOKU	Up	Silencing	[12]
DN34106_c0_g1_i6	NLR family CARD domain-containing protein 3	Up	Ubiquitination, mTOR pathway	[9]
DN38412_c0_g4_i1	BTB/POZ and MATH domain-containing protein 2	Down	Ubiquitination	[40]

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
