# Peer review of "Genes Modulating the Increase in Sexuality in the Facultative Diplosporous Grass Eragrostis curvula under Water Stress Conditions"

_genes, 2020, doi:10.3390/genes11090969_

Round 1
Reviewer 1 Report
Review of genes 884693 — Genes modulating the increase of sexuality in the facultative diplosporous grass Eragrostis curvula under water stress conditions by Selva et al.
The authors used a facultatively apomictic cultivar of Eragrostis curvula (Poaceae) to look for the genes responsible for increased sexual embryo sac production under conditions of water deficiency stress, so as to have a more detailed understanding of the regulation of apomixis. The authors grew clones of three individuals either under water deprivation or sufficient water, and compared their genome-wide transcript profile. In addition, they compared the differentially expressed transcripts with those (also DE) between apomictic and sexual plants from a previous study. The authors identified a number of candidate genes involved in the regulation of apomixis.
The paper is generally well-written, clear, and the results will be useful in understanding how apomixis is regulated, and how stress can induce sexuality in apomictic individuals. The results will be highly relevant for a better understanding of the evolution of apomixis from sexual ancestors, and of facultative sexuality in apomictic species.
The paper could be improved in two main aspects. First, the way in which the objectives are presented sets up expectations that are not met: 1) a clear relation between gene expression in flowers and the number of sexual embryo sacs produced; and 2) a detailed model of the regulation of apomixis. The authors could caution the reader on how difficult it could be to achieve these goals right after posing them. While the authors do list the main caveats of the study at the start of the discussion, I was still expecting at some point to see a figure with the model as the authors currently envision the pathways involved in de-repressing sexuality. Since the information gathered does not allow that level of detail, the authors may want to provide some qualifiers regarding the model they can propose (L103).
Second, it seemed straightforward that the study would need a set of non-apomictic plants (or apomictic ones that could not switch to sexual) that could be subjected to stress so that the transcripts that would be differentially expressed due to stress alone (without the induction of sexuality) could be teased out from those that regulate apomixis. The lack of these data leaves an important gap in a study that would otherwise be very well-rounded.
Specific comments and some minor corrections are listed below:
L45: “high-yielding”
L53: The genes could be listed in a table, for easier reading of the text.
L59: Why is it called cyclical? This would suggest that there is an alternation, regardless of stress. If there is an alternation of favourable and unfavourable seasons, then that should be stated.
Replace “… the organism reproduce by apomixis in one season, generally under favourable conditions, and sexual during…” with “reproduction is apomictic in one season, generally under favourable conditions, and sexual…”
L61: Do the interacting organisms cause the stressful conditions? If the abiotic stress alone can induce apomixis (as explained further along the text), do the biotic interactions exacerbate the stress? If that is the case, I suggest the abiotic stress topic is presented before the interaction with the biotic stressors.
L74: There must be some reviews regarding this topic, please check for references.
L79: If this is the first mention of gene ontology, add the acronym (GO) here. (Check other acronyms throughout the text.)
L90: The reader does not necessarily know which of the cited studies are from the authors’ group. Therefore, it will read better if the authors replace “We also” with “Our group”
L94: replace “less” with “fewer”
L98: Spell out RdDM the first time mentioned.
L100: The aim of the paper seems somewhat disconnected from the introduction. What is novel about the approach? What are the gaps in knowledge that the study will address? What did the previous studies miss?
Materials and methods
L105: Why did the authors not have a set of non-apomictic plants? This would give them patterns of DE resulting from water stress without those due to apomixis? Would this information be available already from previous studies in this species?
Results
L203: Replace “a lower number of” with “fewer”. Use “more” on L204 and L205.
Figure 1. No need to repeat the bar colour legend on all three graphs. DE and GO should be spelled out.
L239: This is where the data on gene expression of non-apomictic or non-facultative apomictic plants subjected to stress would be useful.
L259: Not clear: what is the pathway that is highlighted? State that first, then the reasons why it is considered important.
L313: Are “beta-expansin” and “tillers1” supposed to be in italics and / or upper case?
Table 2. The headers for the sections of the table need to match in font - position.
L332: Check for typos.
L362: While the figure for this section shows well the up- and down-regulated transcripts, the authors should briefly explain the results shown in terms of the gene ontologies of said transcripts (if available).
Discussion
L384: Regarding the second consideration, one would have expected the authors to have analyze the transcript profiles of negative controls (non-apomictic plants subjected to water deficiency-stress). Figure 3 shows Venn diagrams for DE genes due to stress only. Why is this not done seq by seq so that the DE responsible for apomixis is clearly shown?
L445: Italics for P. notatum
L451: Period missing
L454: None of the DE transcripts shared between stress and apomixis (from previous data) were regulated in the same direction (up/up or down/down)?
L482: Not clear what the authors mean by “interconnected roads” in reference to the two kinds of hormones. Consider re-writing this part.
L512: This last sentence is too long and unclear.
Author Response
Dear reviewer 1,
Please see the attachment.
Best regards,
Dr. Viviana Echenique

Reviewer 2 Report
Selva et al investigated the role of stress in regulating apomixis. Plants of the facultative apomictic cultivar of the African grass Eragrostis curvula were exposed to different water availability levels (control and stress). They found changes in the percentage of apomictic and sexual embryo sacs and identified different patterns of gene expression in response to water stress. Overall, I found the study very interesting, well justified and just have a few minor comments listed below.
Lines 108-109: could you clarify/confirm whether your experiment consisted of three genetically different individuals or just one single genet (i.e., all clones in the experiment)? perhaps by adding a clarification as the suggestion in brackets.
three plants were divided asexually (tillers) and one clone was assigned to the control treatment and the other one to the stress condition, totalling six plants (three genetically different individuals that were assigned both to control and stress conditions).
Lines 113-115: Why/How the weekly water amount was chosen? What do you mean by a supplementary irrigation? was this to field capacity? or just an additional 50-80 ml per pot
Line 116: Could you indicate when RWC was measured? was this at harvest? or during the experiment?
Lines 120-128: Could you indicate how many samples? how many observations were carried out?
Lines 182-183: Italicize E. curvula
Line 248: Arabidopsis in italics
Figure 1: Could you add a title to Y-axis?
Figure 2: Add title to X-axis
Figure 3: This figure is hard to follow/understand. Legend to figure needs more detail to fully understand what is shown.
Line 398: check grammar – increased
Line 445: Italicize P. notatum
Line 451: Add a full stop after [97]
Author Response
Dear Reviewer 2,
Please see the attachment.
Best regards,
Dr. Viviana Echenique

Reviewer 3 Report
The authors of the manuscript have adequately presented results that demonstrate the effect of water stress on the transcriptional profiling of the facultative diplosporous grass Eragrostis curvula. Moreover, they have discussed the link between stress conditions and the apomixis/sexual switch both in physiological and in transcriptional level. Nevertheless, there is one point that needs attention. The authors suggest in the abstract that "a DNA glycosylase EcROS1-like could be demethylating, thus de-repressing gene or genes involved in the sexuality pathways". Additionally at line 487 they demonstrate that the ROS1 is "overexpressed under stress conditions in the present study". In figure 4, the ROS1-like gene in water stressed plants is upregulated but not with statistical significance. Since the authors' argument relies on the overexpression on that gene, they should repeat the experiment or analyse downstream targets of the ROS1-like gene to show the changes in the pathways. Additionally, their proposed model should be ellaborated.
Author Response
Dear Reviewer 3,
Please see the attachment.
Best regards,
Dr. Viviana Echenique

Round 2
Reviewer 1 Report
I am satisfied with the revisions made by the authors and with their responses to my comments.
Given that at this time the authors do not have the biological material needed to address my second concern (the need of non-apomictic plants so that one could investigate the differentially expressed genes due to stress alone without apomixis), the authors could insert a brief explanation of why they could not include this in their study (as explained in their response) - a short paragraph after L 118 would suffice.
A couple of minor corrections:
L 105: A few words need to be deleted / changed to improve clarity: “This approach was taken based on our previous findings about increases of sexual processes under stress conditions [40] in order to look for common pathways between stress and apomixis that could give clues about the regulation of this intriguing reproductive mode”
L 215: Change “…had a fewer number of flowers…” to “…had fewer flowers…”
L 217: Change “… observe more number of spikelets…” to “… observe more spikelets…”
Author Response
Dear Reviewer,
Please see the attachment.
Kind Regards,
Dr Viviana Echenique

Reviewer 3 Report
In the revised form of the manuscript, the authors made an adequate presentation of the results regarding the "Validation by qRT-PCR" section and the expression profile of the EcROS1-like
gene. The manuscript can be better if two minor points are improved in the revised version:
- Please provide figures with better resolution
- Please check the abbreviations and their order of appearance in text and figures.
Author Response

(The authors gave the same response as above.)
